# Using real-time modelling to inform the 2017 Ebola outbreak response in DR Congo

R. Thompson [1] ✉, W. Hart [1], M. Keita[2,3], I. Fall[4], A. Gueye[2], D. Chamla[2], M. Mossoko[5], S. Ahuka-Mundeke[6], J. Nsio-Mbeta[5], T. Jombart[7] & J. Polonsky [8]

Important policy questions during infections disease outbreaks include: i) How effective are particular interventions?; ii) When can resource-intensive interventions be removed? We used mathematical modelling to address these questions during the 2017 Ebola outbreak in Likati Health Zone, Democratic Republic of the Congo (DRC). Eight cases occurred before 15 May 2017, when the Ebola Response Team (ERT; co-ordinated by the World Health Organisation and DRC Ministry of Health) was deployed to reduce transmission. We used a branching process model to estimate that, pre-ERT arrival, the reproduction number was $R = 1.49$ (95% credible interval (0.67,2.81)). The risk of further cases occurring without the ERT was estimated to be 0.97 (97%). However, no cases materialised, suggesting that the ERT's measures were effective. We also estimated the risk of withdrawing the ERT in real-time. By the actual ERT withdrawal date (2 July 2017), the risk of future cases without the ERT was only 0.01, indicating that the ERT withdrawal decision was safe. We evaluated the sensitivity of our results to the estimated $R$ value and considered different criteria for determining the ERT withdrawal date. This research provides an extensible modelling framework that can be used to guide decisions about when to relax interventions during future outbreaks.

The 2014–16 Ebola virus disease (EVD) epidemic in West Africa was a stark reminder of the need for rapid response to control emerging infectious disease outbreaks[1–4]. In recent years, epidemiological modelling has increasingly been used to track pathogen transmission and to guide public health measures against a range of diseases[5–9]. However, beyond the timely deployment of interventions, identifying the adequate level of outbreak response and understanding how to adjust measures over time are challenging issues[10–14].

An EVD outbreak occurred in 2017 in Likati Health Zone, Democratic Republic of the Congo (DRC)[15–17]. The index case developed symptoms on 27 March 2017. After eight cases had arisen in total, a response team (the Ebola Response Team; ERT) co-ordinated by the

DRC Ministry of Health and the World Health Organisation was deployed and implemented a range of measures, including community engagement and risk communication, contact tracing, active case finding, isolation and treatment of suspected patients, and PCR and serology testing. After the deployment of the ERT, the key question was whether these actions were sufficient to prevent further transmission or whether additional measures were required. Around one month later, when no further cases had occurred, attention turned to when public health measures could be relaxed and the ERT could be withdrawn safely with only a small risk of outbreak resurgence. Efficient removal of control resources, when they are no longer necessary, is essential to reduce economic and social costs[18].

[1]Mathematical Institute, University of Oxford, Oxford, UK. [2]World Health Organization, Regional Office for Africa, Brazzaville, Democratic Republic of the Congo. [3]Institute of Global Health, Faculty of Medicine, University of Geneva, Geneva, Switzerland. [4]Global Neglected Tropical Diseases Programme, World Health Organization, Geneva, Switzerland. [5]Institut National de Santé Publique, Ministry of Public Health, Hygiene and Prevention, Kinshasa, Democratic Republic of the Congo. [6]Institut National de Recherche Biomédicale, Kinshasa, Democratic Republic of the Congo. [7]MRC Centre for Global Infectious Disease Analysis, School of Public Health, Imperial College, London, UK. [8]Geneva Centre of Humanitarian Studies, University of Geneva, Geneva, Switzerland. ✉e-mail: robin.thompson@maths.ox.ac.uk

Mathematical models can be used both to assess the effectiveness of public health measures and to determine when interventions can be lifted safely. For example, Funk et al.[19] analysed data from the beginning of the 2014–16 Ebola epidemic and demonstrated that expansion of the treatment centre and an increase in healthcare-seeking behaviour limited transmission in Lofa County, Liberia. Similarly, a range of studies attempted to unpick the effects of different interventions during the COVID-19 pandemic[20–22]. In terms of deciding when to remove interventions, other modelling analyses have been used to assess the "end-of-outbreak probability": namely, the probability that an outbreak is over and no further cases will occur in future[23–30]. Estimation of the end-of-outbreak probability is useful for informing when outbreaks can be declared over and when interventions can be removed, since this decision-making involves balancing the benefits of relaxing stringent and resource-intensive measures against the risk of additional cases.

In this article, we report mathematical modelling that we undertook during and after the 2017 EVD outbreak in Likati Health Zone, DRC, to address two key questions, specifically: (i) How effective was the ERT at reducing transmission (Fig. 1—Key Question 1)?; (ii) When could the ERT be withdrawn and associated public health measures relaxed without a substantial risk of further cases (Fig. 1—Key Question 2)? We used a branching process outbreak model to infer the level of virus transmission prior to the arrival of the ERT, as quantified by the reproduction number, $R$. Following the deployment of the ERT, each day, we estimated the risk of additional cases occurring if the ERT was withdrawn under the assumption that $R$ reverts to its original value (i.e., the value estimated using data from before the arrival of the ERT) in the absence of the ERT. As we show, this quantity, which we term the "risk of withdrawing the ERT", can be used to assess the effectiveness of the ERT and to guide in-real time when the ERT can be withdrawn according to an acceptable level of risk (to be determined in a context-specific fashion). As well as providing analyses of the 2017 EVD outbreak in Likati Health Zone, our research provides a general epidemiological modelling framework that can be used during future outbreaks to assess interventions and to determine when measures can be relaxed or removed safely.

## Results

### Analysing the outbreak response

To analyse transmission in the absence of the ERT, we began by estimating the value of $R$ using: (i) the incidence data from 27 March to 14 May 2017 and (ii) a distributional estimate of the EVD serial interval (see the "Methods" section and Fig. S1). This range of dates represents the time period from the onset of disease in the index case up to the day before the arrival of the ERT (Fig. 1). In the absence of the ERT, the

median reproduction number estimate was $R = 1.49$ (95% credible interval 0.67–2.81; Fig. 2A).

Following the arrival of the ERT, no further cases occurred in the outbreak. On each day, we estimated the probability that future cases would occur if the ERT had been withdrawn (i.e., the risk of withdrawing the ERT), under the assumption that $R$ reverts to its original value (i.e., the value estimated using the disease incidence time series data from before the arrival of the ERT) when the ERT is withdrawn (Fig. 2B). As described in the "Methods" section (Eq. (5)), in this analysis the risk of withdrawing the ERT was calculated using the full distributional estimate of $R$ in the absence of the ERT (Fig. 2A).

Two things are notable from the estimated risk of withdrawing the ERT shown in Fig. 2B. First, on the day that the ERT arrived, there was a high estimated probability of future cases (0.97) in the absence of the ERT. The fact that no subsequent cases occurred suggests that the ERT was effective at reducing transmission. Second, there was a very small probability of future cases in the absence of the ERT by the actual date on which the ERT was withdrawn (0.01 on 2 July 2017). Policy makers could, therefore, be confident (around 99% sure) that no further cases would occur when the outbreak was declared over and the ERT was withdrawn.

### Using models to guide withdrawal of the ERT

The analysis in Fig. 2 demonstrates that further cases were unlikely by the time that the ERT was withdrawn from the Likati Health Zone (2 July 2017). Such analyses can be used in real-time to guide when interventions can be relaxed or removed. For example, the ERT could theoretically have been withdrawn as soon as the risk of withdrawing the ERT fell below a pre-specified threshold value. While such a threshold was not specified for this outbreak, smaller threshold values should be set when policy makers have a lower tolerance for future cases occurring. In this outbreak, if a threshold value of 0.05 had been chosen (corresponding to a <5% chance of future cases), then the ERT could have been withdrawn on 21 June 2017, whereas if a more risk-averse threshold value of 0.01 had been chosen (corresponding to a <1% chance of future cases), then the ERT could instead have been withdrawn on 3 July 2017, which corresponds very closely to the actual date of ERT withdrawal (2 July 2017). In practice, the outbreak was declared over and the ERT was withdrawn based on the one-size-fits-all rule of declaring EVD outbreaks over after a period of 42 days without cases (following the recovery or burial of the previous case)[31].

While the results in Fig. 2B are based on the full distributional estimate of $R$, other assumptions are possible. For example, if a policy maker wishes to be risk averse, they may choose to base their decision-making on analyses conducted with a high value of $R$ rather than the

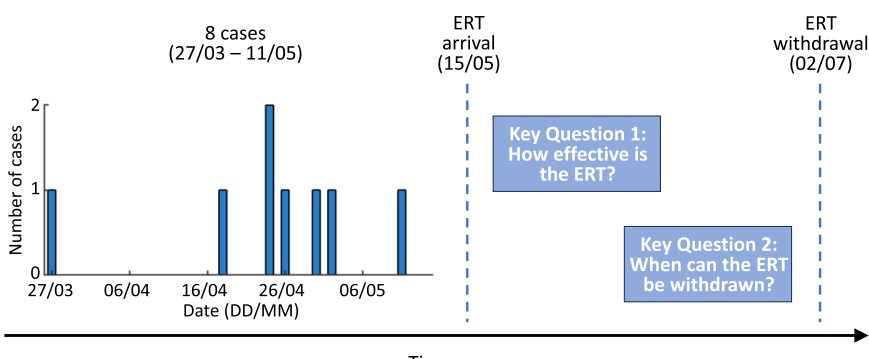

**Fig. 1 | Schematic illustrating the key questions that mathematical modelling was used to address during the 2017 EVD outbreak in Likati Health Zone, DRC.** Following eight EVD cases occurring between 27 March and 11 May 2017, the ERT was deployed on 15 May 2017. Mathematical modelling was used to assess: (i) The effectiveness of the ERT at reducing transmission (Key Question 1), and; (ii) The risk of withdrawing the ERT, quantified in terms of the probability that further cases would occur if the ERT was withdrawn (Key Question 2; this quantity was evaluated every day until the ERT was withdrawn on 2 July 2017). In this figure, and in all subsequent figures in the main text, dates are expressed in DD/MM format (all in 2017).

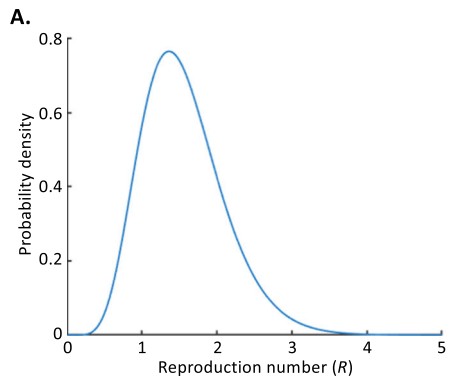
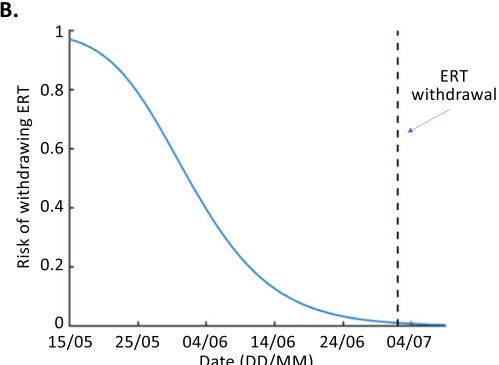

**Fig. 2 | Assessing the effectiveness of the ERT and the risk of withdrawing the ERT. A** The estimated value of $R$ prior to the arrival of the ERT (calculated using Eq. (3)). **B** The risk of withdrawing the ERT (blue, calculated each day using Eq. (5); i.e., the probability of future cases occurring if the ERT is withdrawn on each date on the $x$-axis, based on the distributional estimate of $R$ in panel **A**) and the actual date of withdrawal of the ERT (black dashed). In panel **B**, the first date shown is the ERT deployment date (15 May 2017).

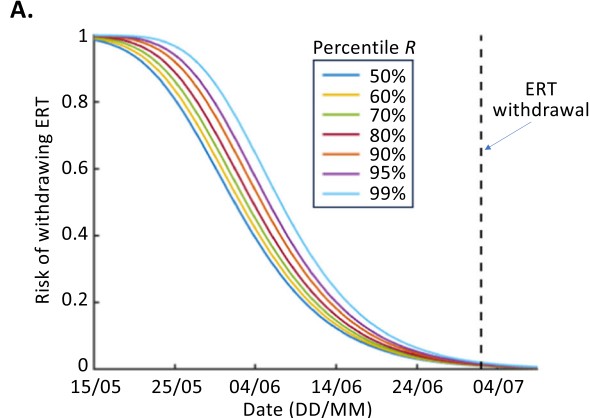
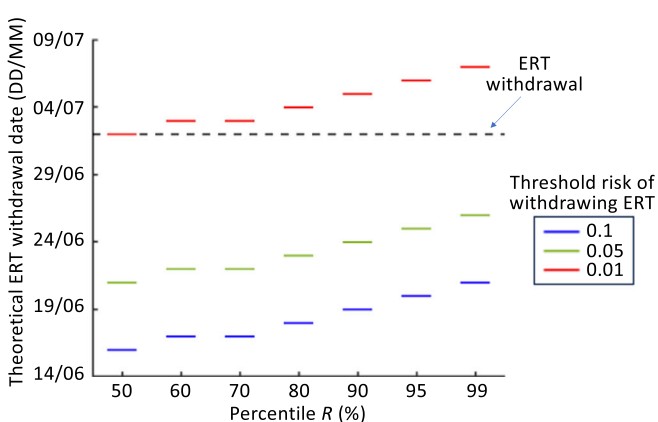

**Fig. 3 | Determining when to withdraw the ERT for different assumed values of $R$ and different risk tolerance levels. A** The risk of withdrawing the ERT each day (Eq. (4)) for different percentile values of the distributional estimate of $R$ (see Fig. 2A) and the actual date of withdrawal of the ERT (black dashed). **B** The date on which the ERT could theoretically have been withdrawn, if the ERT was withdrawn as soon as the risk of withdrawing the ERT fell below different threshold values (0.1−blue; 0.05−green; 0.01−red). A threshold value of 0.01 corresponds to (at most) a 1% chance of cases arising following ERT withdrawal. Results are shown for different percentile values from the distributional estimate of $R$ (see Fig. 2A). In both panels, the percentile $R$ values shown correspond to: 50% − $R = 1.49$; 60% − $R = 1.63$; 70% − $R = 1.79$; 80% − $R = 1.99$; 90% − $R = 2.29$; 95% − $R = 2.56$; 99% − $R = 3.11$.

full distributional estimate of $R$. For example, if the analysis in Fig. 2B is repeated but with $R = 2.56$ (corresponding to the 95th percentile estimate of $R$ in Fig. 2A), then the result shown in purple in Fig. 3A is obtained. Similar results but for different percentile estimates of $R$ from the distributional estimate in Fig. 2A are also shown in Fig. 3A (these results correspond to applying Eq. (4) with different values of $R$).

We find that the higher the assumed value of $R$, the longer it would have been necessary to wait before the inferred risk of future cases in the absence of the ERT was low enough for the ERT to be withdrawn (Fig. 3B). Similarly, if the ERT can only be withdrawn when the risk of future cases reduces below a pre-specified threshold value, then as expected a lower value of this threshold would require policy makers to wait longer before withdrawing the ERT. For example, if the median estimate of $R$ was used, and interventions were removed as soon as the risk of future cases fell below 0.1 (i.e., 10%), then the ERT could have been withdrawn on 16 June 2017. If, instead, the 95th percentile estimate of $R$ was used, and interventions were removed as soon as the risk of future cases fell below 0.01 (i.e., 1%), then the ERT withdrawal date would have been 6 July 2017.

### Extensibility of our modelling approach
To demonstrate how our modelling approach can be extended for use in different settings, we conducted a range of Supplementary Analyses.

In the branching process model used to generate the results shown in the main text (Figs. 2 and 3), we assumed that the number of cases each day was drawn from a Poisson distribution (see the "Methods" section). While this assumption is made frequently in such models[32–34], other probability distributions could be used. For example, use of a negative binomial distribution with a low value of the dispersion parameter, $k$, allows the possibility of super-spreading events to be accounted for[35]. In Supplementary Analysis 1 and Fig. S2, we reproduced the results shown in Fig. 2 but under this alternative transmission model. We found that our main conclusions were unchanged. First, soon after the ERT was deployed, there was a high risk of future cases in the absence of the ERT (0.97 under both the assumption that $k = 0.2$ and under the alternative assumption that $k = 10$). The fact that no cases went on to occur, despite the high risk of additional cases in the absence of the ERT, suggests that the ERT was effective. Second, by the time that the ERT was withdrawn, the risk of ERT withdrawal was estimated to be low (0.02 when $k = 0.2$ and 0.01 when $k = 10$), indicating that the decision to withdraw the ERT was safe (Fig. S2B).

To demonstrate the generalisability of our modelling approach, we also applied it to data from a second larger outbreak of EVD in which the ERT was deployed. Specifically, in Supplementary Analysis 2 and Fig. S3, we estimated the risk of ERT withdrawal each day for an

EVD outbreak with 54 cases that occurred in Équateur Province of DRC in 2018. Because this was a larger outbreak than the one analysed in the main text, contact tracing allowed sufficient data to be recorded for the serial interval distribution to be estimated for this specific outbreak (rather than the general EVD serial interval estimate that we used in our main analyses—see the "Methods" section). Again, in the 2018 EVD outbreak, our estimates suggest that the ERT was only withdrawn when it was safe to do so (Fig. S3E).

In our main analyses, we assumed that all cases were recorded. While this was a plausible assumption for the 2017 EVD outbreak in Likati Health Zone due to its small size and the extensive case finding that was undertaken, for larger EVD outbreaks underreporting of cases is common[36]. In Supplementary Analysis 3 and Fig. S4, we, therefore, extended our analysis of the larger 2018 EVD outbreak from Supplementary Analysis 2 to consider the effect of underreporting. This enabled us to demonstrate how unreported cases can be accounted for when estimating the risk of withdrawing the ERT (Fig. S4). As expected, under the assumption of a higher number of unreported cases, the estimated risk of withdrawing the ERT increased. By assuming a larger extent of underreporting, more risk-averse decisions about when to withdraw the ERT can be made.

## Discussion

Assessing the effectiveness of control measures and determining when interventions can be removed safely is essential during infectious disease outbreaks[18,37]. Relaxing interventions too soon presents a risk of a resurgence in cases, but leaving measures in place for longer than necessary is costly. In addition, interventions are, at best, inconvenient and, at worst, highly damaging, to the local population. In this study, we have presented analyses of the 2017 outbreak of EVD in Likati Health Zone, DRC, in which we used mathematical modelling to: (i) assess the effectiveness of the ERT for reducing transmission and (ii) estimate the risk of withdrawing the ERT each day (i.e., the risk that further cases would occur if the ERT was withdrawn) to guide decision making.

The available quantitative evidence suggests that the ERT was effective at reducing transmission during this outbreak. Based on transmission prior to the arrival of the ERT, we estimated that, had the ERT not been deployed, the risk of additional cases occurring was 0.97 (Fig. 2B). Since no cases went on to occur, our analysis suggests that the ERT was effective. The ERT was then in place from 15 May to 2 July 2017. By the time the outbreak was declared over and the ERT was withdrawn, the risk of further cases had fallen to 0.01. While it was only possible to infer that the ERT was effective after the fact (given the additional information that no cases occurred following its arrival), we were able to estimate the risk of withdrawing the ERT in real-time by inferring the probability that additional cases would occur if it was removed, based on the level of transmission observed prior to the ERT's arrival.

Policy makers could choose to remove control interventions as soon as the risk of future cases falls below a context-dependent threshold value. Lower values of this threshold correspond to a more risk-averse strategy (interventions must be maintained for longer, but with a lower risk of subsequent cases occurring, compared to when a higher threshold value is used). If a threshold value of 0.05 had been used to determine when the ERT could be withdrawn in the 2017 EVD outbreak in Likati Health Zone, then the ERT could have been withdrawn on 21 June 2017 (11 days prior to the actual date of withdrawal of the ERT; Fig. 2B).

Our main results relating to the risk of withdrawing the ERT were based on our full distributional estimate of the reproduction number, $R$, in the absence of the ERT (Fig. 2). We then explored how our results differed for individual values of $R$ within the distributional estimate (Fig. 3). Basing policy decisions on a high percentile value of $R$ would be a more risk averse choice than using the full distributional estimate of $R$. We found that, if the ERT is withdrawn when the risk of future cases falls below 0.01, then using the 99th percentile value of $R$ to

guide decision making would have required the ERT to be deployed for longer than using the 50th percentile value of $R$ (until 7 July 2017, rather than 2 July 2017; Fig. 3B). When the full distributional $R$ estimate was used, the corresponding withdrawal date was 3 July 2017 (Fig. 2B).

The fact that higher assumed values of $R$ require policy makers to wait longer before safe withdrawal of the ERT would have been deemed to be possible highlights the fact that considering uncertainty in the value of $R$ in end-of-outbreak analyses is essential. Basing decisions on mean or median estimates of $R$ alone, without considering uncertainty, could lead to interventions being removed on dates that are not supported by all available quantitative evidence. We note that the results shown in Fig. 2B are based on the full distributional estimate of $R$ derived directly from the disease incidence time series data and, therefore, account rigorously for uncertainty in the precise value of $R$.

In our analyses of the 2017 Likati Health Zone EVD outbreak, we attributed the fact that no cases occurred after the arrival of the ERT, despite the high estimated probability of future cases in the absence of the ERT, to the ERT's activities. While we contend that this interpretation is supported by the available quantitative evidence (as noted above, we estimated the risk of additional cases occurring had the ERT not been deployed to be 0.97, yet no further cases arose), other factors may have contributed to preventing transmission. For example, behavioural changes may have occurred in the local population to reduce viral spread even without the ERT being deployed in response to a growing awareness of the outbreak.

As with any epidemiological modelling study, the results presented here are based on simplifying assumptions. An assumption of the transmission model underlying most of our analyses (Figs. 2, 3, S3 and S4) is that the number of cases occurring each day is drawn from a Poisson distribution. While this assumption is common to many renewal equation models[32–34], consideration of other distributions is possible, including accounting for the possibility of super-spreading events[23,38]. For a fixed expected number of events on a particular day, more over-dispersed distributions would be more likely to lead to zero cases on that day, but with a higher chance of a large number of cases occurring on that day reflective of a super-spreading event. To relax the assumption of a Poisson-distributed number of cases each day, we reproduced the results shown in Fig. 2 but instead assumed that the daily number of cases is drawn from a negative binomial distribution, allowing for the possibility of super-spreading events (Supplementary Analysis 1 and Fig. S2). In a scenario in which the ERT is withdrawn as soon as the inferred risk of withdrawing the ERT reaches a pre-specified low value, we found that the ERT withdrawal date was not sensitive to our choice of dispersion parameter in the negative binomial distribution.

Another key assumption underlying our main analyses is that all cases were detected. As noted in the "Results" section, this may have been a reasonable assumption for the 2017 EVD outbreak in Likati Health Zone. However, underreporting of cases is common, not only for EVD but also for a range of other pathogens[36,39,40]. We therefore conducted supplementary analyses of a different (much larger) EVD outbreak (Supplementary Analysis 2; Fig. S3), including considering the effect of unreported cases (Supplementary Analysis 3; Fig. S4), showing that the risk of withdrawing the ERT is higher when underreporting is accounted for (Fig. S4B).

Despite the simplifications in our epidemiological model, our results strongly suggest that the measures implemented by the ERT were effective at reducing transmission during the 2017 EVD outbreak in Likati Health Zone, DRC, and that the ERT was only withdrawn when it was safe to do so. The timely application of effective control measures is extremely important during infectious disease outbreaks. Similarly, the relaxation of interventions as soon as the risk of future cases is sufficiently low allows limited control resources to be conserved. The research conducted here provides a quantitative framework that can be used to guide this decision-making in real-time during future infectious disease outbreaks.

## Methods

### Outbreak data

The 2017 EVD outbreak in Likati Health Zone, DRC, comprised eight cases (five confirmed cases and three probable cases[15,16]) occurring between 27 March and 11 May 2017. The ERT arrived in Likati on 15 May 2017, and no cases arose subsequently. A disease incidence time series was made available in real-time and consisted of symptom onset dates for all reported cases (Fig. 1).

### Epidemiological model

Pathogen transmission in the absence of the ERT was modelled using a renewal equation. In the version of the model used in our main analyses, the number of cases, $I_t$, occurring on day $t$ is drawn from a Poisson distribution with mean

$$\mathbb{E}\left(I_t|R\right) = R\sum_{s=1}^{t-1} I_{t-s}w_s. \tag{1}$$

In this expression, $R$ is the reproduction number (the expected number of secondary cases generated by each case over the course of their infectious period). The notation $w_s$ represents the probability that the (discrete) serial interval takes the value $s$ days (i.e., the probability that the interval between an infector and infectee appearing in the disease incidence time series is $s$ days).

We assumed that the continuous serial interval distribution for EVD is a gamma distribution with a mean of 15.3 days and a standard deviation of 9.3 days[41] (we considered an alternative serial interval distribution in Supplementary Analyses 2 and 3). We then discretised this distribution to obtain $w_s$ (for $s = 1, 2, \ldots$) using the method described in web appendix 11 of the Supplementary Data of the article by Cori et al.[32]. Specifically, if the probability density function of the continuous serial interval distribution is denoted by $g(.)$, then

$$w_s = \frac{1}{N}\int_{s-1}^{s+1}(1 - |u - s|)g(u)\mathrm{d}u, \tag{2}$$

in which $N$ is a normalising constant so that the sequence of values $\{w_s\}_{s=1}^{\infty}$ represents a valid probability mass function. The resulting discretised serial interval distribution is shown in Fig. S1.

### Inference of R

For the purpose of our analyses of the 2017 EVD outbreak in Likati Health Zone, we labelled the date of the first case (27 March 2017) as $t = 1$. The ERT then arrived on day $t = 50$. We used the incidence data up to the day before the arrival of the ERT to estimate $R$ in the absence of the ERT. Specifically, we assumed that the index case arose as a result of transmission from outside the local population (e.g., infection from an animal reservoir), and then calculated the normalised likelihood of the incidence data observed following the index case as a result of local transmission up to (and including) day $t = 49$,

$$L(R) = \frac{1}{M_1}\prod_{t=2}^{49}\frac{\left(R\sum_{s=1}^{t-1}I_{t-s}w_s\right)^{I_t}\exp\left(-R\sum_{s=1}^{t-1}I_{t-s}w_s\right)}{I_t!},$$
$$= \frac{1}{M_2}R^{\left(\sum_{t=2}^{49}I_t\right)}\exp\left(-R\sum_{t=2}^{49}\sum_{s=1}^{t-1}I_{t-s}w_s\right). \tag{3}$$

In these equations, the variables $M_1$ and $M_2$ are normalising constants so that $L(R)$ represents a valid probability distribution (this is equivalent to the posterior estimate for $R$ assuming a uniform prior). This corresponds to a gamma distribution with shape parameter $\alpha = 1 + \sum_{t=2}^{49}I_t$ and rate parameter $\beta = \sum_{t=2}^{49}\sum_{s=1}^{t-1}I_{t-s}w_s = \sum_{s=1}^{48}I_{49-s}F_s$, in which $F_s$ represents the cumulative distribution function of the

discretised serial interval distribution (i.e., the probability that the serial interval takes a value less than or equal to $s$ days). The resulting distribution is shown in Fig. 2A. Additional description about the approach used to estimate $R$ is provided in the Supplementary Information (Supplementary Text 1).

### The risk of withdrawing the Ebola Response Team

Following the arrival of the ERT, no further cases occurred in the 2017 EVD outbreak (this was not true for the larger outbreak considered in Supplementary Analyses 2 and 3). However, deployment of the ERT is costly and the interventions implemented by the ERT can be restrictive to the local population. There are, therefore, incentives to withdraw the ERT and relax or remove associated interventions as quickly as possible when the risk of a resurgence of cases is sufficiently low. We used the distributional estimate of $R$ in the absence of the ERT (Fig. 2A) to determine the risk of future cases if the ERT is withdrawn on day $t$.

For a fixed value of $R$, the probability of future cases occurring at any time from day $t$ onwards, if the ERT is withdrawn on day $t$, is given by

Risk of withdrawing ERT on day $t$

$$= 1 - \text{Prob}\,(\text{no cases from day }t\text{ onwards}|R),$$
$$= 1 - \prod_{j=t}^{\infty}\exp\left(-R\sum_{s=1}^{j-1}I_{j-s}w_s\right),$$
$$= 1 - \exp(-R\gamma(t)). \tag{4}$$

Here, we define $\gamma(t) = \sum_{j=t}^{\infty}\sum_{s=1}^{j-1}I_{j-s}w_s = \sum_{s=1}^{t-1}I_{t-s}(1 - F_{s-1})$. In these expressions, since we are calculating the probability that no cases occur from day $t$ onwards, we set $I_{j-s} = 0$ whenever $j - s \geq t$.

Equation (4) can be used to calculate the risk of withdrawing the ERT when the value of $R$ is known or a single value of $R$ is assumed. To account for uncertainty in the value of $R$, a distributional estimate of $R$ can be incorporated into calculations of this risk using the expression

Risk of withdrawing ERT on day $t$

$$= 1 - \int_0^{\infty}\text{Prob}\,(\text{no cases from day }t\text{ onwards}|R)L(R)\mathrm{d}R,$$
$$= 1 - \int_0^{\infty}\exp(-R\gamma(t))L(R)\mathrm{d}R,$$
$$= 1 - \left(\frac{\beta}{\beta + \gamma(t)}\right)^{\alpha}. \tag{5}$$

Here, $\alpha$ and $\beta$ are the shape and rate parameters of the likelihood, $L(R)$, as given above.

### Ethical approval

An ethics application was submitted by co-author Dr M Keita to allow continued access to and use of multiple Ebola datasets for research purposes. This application was approved by the ethics committee at the Kinshasa School of Public Health, DRC (approval number ESP/CE/03/2021).

### Reporting summary

Further information on research design is available in the Nature Portfolio Reporting Summary linked to this article.

## Data availability

The data used in our analyses are available within the relevant code in the following Github repository: https://github.com/robin-thompson/EbolaReponseTeam[42]. The disease incidence time series from the 2017 EVD outbreak in Likati Health Zone, DRC, has been published previously and there are no restrictions on its use.

## Code availability

The computing code used to perform the analyses in this article is available in the following GitHub repository: https://github.com/robin-thompson/EbolaReponseTeam[42]. All code was written in Matlab (compatible with version 2022a).

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

## Acknowledgements

We acknowledge the support of the JUNIPER partnership, funded by MRC (grant number MR/X018598/1; R.T.). Thanks to members of the Wolfson Centre for Mathematical Biology at the University of Oxford for helpful discussions about this research.

## Author contributions

R.T.: conceptualisation, methodology, formal analysis, investigation, visualisation, validation, writing—original draft, writing—review and editing. W.H.: methodology, validation, writing—review and editing. M.K.: fieldwork, writing—review and editing. I.F.: fieldwork, writing—review and editing. A.G.: fieldwork, writing—review and editing. D.C.: fieldwork, writing—review and editing. M.M.: fieldwork, writing—review and editing. S.A.-M.: fieldwork, writing—review and editing. J.N.-M.: fieldwork, writing—review and editing. T.J.: conceptualisation, methodology, writing—review and editing. J.P.: conceptualisation, fieldwork, methodology, writing—review and editing.

## Competing interests

The authors declare no competing interests.
