## [Peer Review File · Nature Communications]

Using real-time modelling to inform the 2017 Ebola outbreak response in DR CongoREVIEWER COMMENTS

Reviewer #1 (Remarks to the Author):

Review of NCOMMS-24-08812

Understanding when costly measures such as the ERT need to be put in place, and when they can be removed, is obviously an important question. Retrospective analyses can be an important tool to answer that question.

The authors use renewal model to estimate probability of future cases if intervention (ERT) is removed based on distribution of R prior to the intervention. The analysis thus assumes that if the ERT is withdrawn then R will revert to the baseline value estimated prior to the intervention. Sensitivity of the method to higher values of R is explored.

I think the framework developed is potentially useful, but I am puzzled by performance and interpretation given the small number of cases and need more information to evaluate. Assuming the method is reliable, I also have some concerns about inferring the efficacy of the ERT based on a time series of 8 total cases.

MAJOR COMMENTS

1. The authors argue that ERT was likely effective at reducing transmission because, while median estimate of R was relatively high prior to ERT (median 1.49, 95% CI 0.67-2.81), there were no further cases after the ERT deployed. In my view it is not appropriate to infer causality here. Granting the high median R estimate prior to deploying the ERT was accurate (please see below), other factors besides the ERT could still have played a significant role in stopping cases. Indeed, there is not strong basis from this analysis to reject the null hypothesis that the ERT was entirely ineffective. In particular, to be conservative about the impact of the ERT, is it not possible that the outbreak faded out in whole or in part due to demographic stochasticity, given the relatively small number of cases?

2. The analysis uses a very small dataset of 8 cases. As someone who regularly estimates R from incidence time series using related methods (Cori et al) I am surprised that such a small dataset could yield such a precise estimate of R as shown in Fig 2A. I would like to see much more information about the estimation process provided in the Supplemental.

MINOR COMMENTS

The writing in the abstract needs improvement. Two examples are that i) is very nonspecific: how are effective are *particular* public health interventions is the more salient question. Also, the Ebola Response Team is not defined.

Reviewer #2 (Remarks to the Author):

The paper sheds interesting light on incorporating the renewal equation to estimate the effectiveness of public health interventions and how long before the interventions can be safely removed, using the Ebola data from the DRC. The methodology employs an already established renewal equation adapted to understand when and for how long ERT needs to be on ground and has been explained. The results support the claims made in the paper. Here are some comments that authors might want to take into consideration:

1. How does the model take into account the imported cases, since the index case would most likely be seeded or imported (import-to-local transmission)
2. How does the heterogeneity in the generation time distribution distort estimates of the reproduction number and in turn the timing of the implementation of the outbreak response teams.
3. How can this method be adapted in case of a super-spreading event.

4. Would the results differ if cases are used by the dates of symptom onset rather than by the date of case confirmation.
5. How would the results differ if the ERT was deployed before the detection of cases?

Reviewer #2 (Remarks on code availability):

The code is annotated and in detail. It looks reproducible.

Reviewer #3 (Remarks to the Author):

The authors have presented a nice conceptual framework for addressing the end of NPI-based interventions. While I appreciate the simplicity and elegance of their analysis, I am concerned that 1), at present, it does not go far enough to address the practical challenges of implementing such an analysis outside the narrow framing of this specific outbreak, and 2) the current example (a very small outbreak with a very effective intervention) is too idiosyncratic to illustrate the range of outcomes for this analysis.

With respect to the first:

- I would like to see the authors specifically address the issue of under-reporting or cryptic transmission. The analysis appears to be quite dependent on the assumption that all transmission events are seen. That may be a reasonable assumption for this particular outbreak, but is likely not the case for larger outbreaks or non-Ebola outbreaks.
- the authors discuss the assumption that the number of cases per day is Poisson distributed as a limitation. This seems a strong assumption that should be addressed here. It would help to see Figures 2 and 3 with alternative assumptions (I presume that alternative assumptions will impact both the estimate of R (Figure 2a) and the probability of more cases upon removal).
- I would like to see an analysis of the sensitivity of the results to e.g. a missed case at the time of the decision to remove the intervention. Especially given the small size of this outbreak, I would think that missing a single case might be impactful (but would be happy to be proven wrong). This gets at the interaction between this particular decision framework and the strength of surveillance. For example, I could imagine a result where this is a very robust decision framework IF one is very confident in the surveillance (numbers of cases and timing of cases), but one might want to move to a more conservative risk threshold if you're less confident of the surveillance.

With respect to the second:

- I am concerned that such a small outbreak may be idiosyncratic and misleading. I'd recommend that the authors consider either a simulation study based on rates estimated here (perhaps with alternative assumptions about the offspring distribution for the branching process), or an analysis of a larger outbreak where the impact of the intervention is less clear (e.g. cases continue after the intervention). I would prefer the latter as it would help to argue for the generality of this approach.
- a challenge of the simulation approach is that the lack of onward cases after the introduction of the ERT implies that there is large uncertainty in the effectiveness of the ERT; e.g. did it reduce R to 0, less than 1, or greater than 1 and fadeout was a random occurrence? The statement on L147-149 implies that the authors assume that no new case arise after the ERT, again this seems a strong and limiting assumption. The serial interval distribution that the authors assume implies that there is at least a non-trivial probability that transmissions events at the approximate time of the ERT might not even be observed for several weeks. It is interesting to note that the longer the ERT, the more certainty one has in the absence of new transmission; but again, this interacts with the assumption about the effectiveness of the ERT. One might not feel confident in the assumption of effectiveness 1 day after the ERT, but might feel very confident by the 5th of April.

Finally, it strikes me that the assumption of an explicit distribution of serial interval is a very strong assumption that limits this analysis to other Ebola outbreaks, or outbreaks with a well defined serial interval. If the serial interval is not well known, it is not clear that it could be well estimated from such a small outbreak. However, for a larger outbreak, this could also be estimated and the

entire analysis self contained to the specific outbreak at hand. I realize that practical decision-making often requires making such assumptions, but the dependence of these results on this assumption and the limitation of the direct application of this framework to settings with such a well characterized distribution should be addressed here.

Ultimately, I think the authors have presented a very interesting and clean framework for addressing the problem of when to declare the end of NPIs for a small Ebola outbreak where NPIs have been highly effective. That is an achievement by itself, but not one that I think rises to the level of generality that I would expect of Nature Communications. I think, with substantial further analysis, this work could be broadened to illustrate the extent of its generalizability.

Reviewer #4 (Remarks to the Author):

this is a strong manuscript and novel use of modeling to show a positive public health impact. the writing was clear and i don't see need for edits.

We are grateful to the reviewers for their comments that have helped us to improve our manuscript.

Please note: In the material below, whenever we refer to line numbers in the revised manuscript, these refer to the version of the manuscript with “track changes”. We have uploaded two versions of the revised manuscript – one with, and one without, track changes.

Reviewer 1

Understanding when costly measures such as the ERT need to be put in place, and when they can be removed, is obviously an important question. Retrospective analyses can be an important tool to answer that question.

The authors use renewal model to estimate probability of future cases if intervention (ERT) is removed based on distribution of R prior to the intervention. The analysis thus assumes that if the ERT is withdrawn then R will revert to the baseline value estimated prior to the intervention. Sensitivity of the method to higher values of R is explored.

I think the framework developed is potentially useful, but I am puzzled by performance and interpretation given the small number of cases and need more information to evaluate. Assuming the method is reliable, I also have some concerns about inferring the efficacy of the ERT based on a time series of 8 total cases.

***Response:** We agree with the reviewer that the topic addressed in our manuscript is an important one, and thank them for their comments. We address these comments in more detail below.*

MAJOR COMMENTS

1. The authors argue that ERT was likely effective at reducing transmission because, while median estimate of R was relatively high prior to ERT (median 1.49, 95% CI 0.67-2.81), there were no further cases after the ERT deployed. In my view it is not appropriate to infer causality here. Granting the high median R estimate prior to deploying the ERT was accurate (please see below), other factors besides the ERT could still have played a significant role in stopping cases. Indeed, there is not strong basis from this analysis to reject the null hypothesis that the ERT was entirely ineffective. In particular, to be conservative about the impact of the ERT, is it not possible that the outbreak faded out in whole or in part due to demographic stochasticity, given the relatively small number of cases?

***Response:** Thank you very much for this point. We agree with the reviewer that it cannot be inferred with certainty that the ERT was responsible for the lack of additional cases after its arrival. Other factors may also have played a role (e.g. increased public awareness of the outbreak reducing the amount of risky behaviour in the host population). However, under the assumed transmission model, demographic*

stochasticity is highly unlikely to be responsible for the lack of further cases. Based on our analysis, the risk that future cases occurred had the ERT not been deployed was 0.97, meaning that the probability of no future cases (accounting for stochasticity) was only 0.03. Consequently, under the assumptions of the transmission model (R reverting to its pre-ERT value following the withdrawal of the ERT), further cases were highly likely in the absence of the ERT. The fact that no cases occurred despite the near certainty that additional cases would occur without the ERT suggests that the ERT was effective. Nonetheless, we agree with the reviewer that other factors may also have contributed to the apparent success of the ERT; we have provided further details about this in the revised manuscript (lines 809-838). Where appropriate, we have also ensured that the wording in our revised manuscript states that our quantitative results “suggest” that the ERT was effective at reducing transmission, as opposed to stating causality with certainty.

2. The analysis uses a very small dataset of 8 cases. As someone who regularly estimates R from incidence time series using related methods (Cori et al) I am surprised that such a small dataset could yield such a precise estimate of R as shown in Fig 2A. I would like to see much more information about the estimation process provided in the Supplemental.

Response: *This is a really interesting observation. The main difference between our approach and methods used to infer the time varying reproduction number is that we assume a constant value of R prior to the arrival of the ERT. In contrast, when approaches such as those developed by Cori et al. are used, the focus is to infer temporal variations in the time varying reproduction number, i.e. inference of multiple successive values of R_t . Understanding temporal changes in R_t would almost certainly be impossible for such a small dataset (as the reviewer describes, this would require more cases occurring every day for precise estimates to be obtained). In contrast, under the assumption of a single value of R before the arrival of the ERT, precise estimates can be obtained even assuming a non-informative prior (we used a uniform prior in our analysis). This is because information about the value of R in the absence of the ERT accumulates over all days before the arrival of the ERT. We have provided further details about the fitting in the revised Supplementary Information (Supplementary Text 1, which follows the Supplementary Analyses), including explaining how informative priors for R could be incorporated into our analyses (as noted above, in our work we use uninformative priors). We have also explained in Supplementary Text 1 why precise inference of R is possible here (whereas precise estimation of the time-varying reproduction number would have been impossible).*

MINOR COMMENTS

The writing in the abstract needs improvement. Two examples are that i) is very nonspecific: how are effective are *particular* public health interventions is the more salient question. Also, the Ebola Response Team is not defined.

Response: *Thank you for pointing out that the Abstract could have been written more*

carefully. We have made the changes suggested by the reviewer in the revised manuscript and proof-read the Abstract carefully. We have also condensed it, to ensure that it fits within the journal's Abstract word limit.

Reviewer 2

The paper sheds interesting light on incorporating the renewal equation to estimate the effectiveness of public health interventions and how long before the interventions can be safely removed, using the Ebola data from the DRC. The methodology employs an already established renewal equation adapted to understand when and for how long ERT needs to be on ground and has been explained. The results support the claims made in the paper.

Response: *Thank you for these positive comments. We have addressed your questions below.*

Here are some comments that authors might want to take into consideration:

1. How does the model take into account the imported cases, since the index case would most likely be seeded or imported (import-to-local transmission).

Response: *The reviewer is correct that the index case in this outbreak was most likely imported from elsewhere. In our analyses, we assumed that the index case was an imported one and that all further cases arose as a result of local transmission. This is accounted for in the model by estimating the value of R based on infected cases occurring from time $t=2$ onwards (i.e., following the index case, which occurred at $t=1$; please see equation (2) in the manuscript in which the product/summation starts from $t=2$). We have now emphasised this key point in the revised manuscript (lines 904-907).*

2. How does the heterogeneity in the generation time distribution distort estimates of the reproduction number and in turn the timing of the implementation of the outbreak response teams.

Response: *In the new Supplementary Analysis 2, we have conducted a novel analysis in which we infer the reproduction number and risk of withdrawing the ERT using data from a larger outbreak of EVD. Because we were considering a larger outbreak, we were able to estimate the serial interval distribution using data from that outbreak, and compare our modelling results to analogous results if instead the original (not outbreak-specific) serial interval had been used. We found that differences between the two serial interval distributions affected the precise estimated values of both R and the risk of withdrawing the ERT (Fig S3D,E), although the general pattern remained unchanged (the longer the period elapsed following the end of the outbreak, the lower the risk of withdrawing the ERT). Differences in the exact quantitative estimates highlight the importance of using outbreak-specific serial interval estimates when they are available, as we have now highlighted at the end of Supplementary Analysis 2.*

3. How can this method be adapted in case of a super-spreading event.

Response: *Thank you for prompting us to consider the possibility that super-spreading events may occur. To account for this, we have conducted an entirely new analysis in which we relax the assumption that a Poisson distributed number of cases occur every day. In this new supplementary analysis (Supplementary Analysis 1 and Fig S2), we have repeated our original analysis but instead assumed that a negative binomial number of cases occur each day. A negative binomial distribution with a small value of the dispersion parameter, k , corresponds to a higher chance of superspreading events arising than when a higher value of k is used. When $k < 1$, so that super-spreading is relatively likely, we find that there is generally a higher risk of withdrawing the ERT (Fig S2B). However, by the time that the risk of withdrawing the ERT has fallen to low values, so that policy makers may consider withdrawing the ERT, we find that our results look similar for a range of values of k (Fig S2B). This demonstrates the robustness of our results in terms of practical decision making to whether or not super-spreading is accounted for in the underlying transmission model. We describe this in Supplementary Analysis 1, and highlight these results in the main text (lines 720-733 and 840-853).*

4. Would the results differ if cases are used by the dates of symptom onset rather than by the date of case confirmation.

Response: *In fact, the data that we used to assess the risk of removing the ERT were the symptom onset dates. Had we instead used case confirmation dates, some cases would have appeared later, leading to a higher estimated risk of removing the ERT around the time that the ERT was actually withdrawn. However, since the distribution characterising the time period between cases that we used was an estimated serial interval distribution (representing the time between successive symptom onset times in a transmission chain), using symptom onset dates (where available) as we have done is the most appropriate choice. We have now stated explicitly in the revised manuscript that the data underlying our analyses were symptom onset dates (please see lines 875-877 of the revised manuscript).*

5. How would the results differ if the ERT was deployed before the detection of cases?

Response: *If the ERT was deployed before any cases were detected, then it would be impossible to assess the value of R in the absence of the ERT. As such, it would be challenging to make sensible projections regarding the risk of cases if the ERT was withdrawn (other than to say that, if cases were then occurring with the ERT present, then cases would also be highly likely without it). During future outbreaks, we recommend that the ERT is deployed quickly to limit transmission. However, since in practice the ERT is deployed reactively, this would not be before any cases are detected. As a result, we contend that our modelling framework would remain useful for public health decision making.*

The code is annotated and in detail. It looks reproducible.

Response: *Thank you for noting the reproducibility of our research.*

Reviewer 3

The authors have presented a nice conceptual framework for addressing the end of NPI-based interventions. While I appreciate the simplicity and elegance of their analysis, I am concerned that 1), at present, it does not go far enough to address the practical challenges of implementing such an analysis outside the narrow framing of this specific outbreak, and 2) the current example (a very small outbreak with a very effective intervention) is too idiosyncratic to illustrate the range of outcomes for this analysis.

Response: *Thank you for these very important points. We have addressed these comments in detail below, including conducting a range of new supplementary analyses.*

With respect to the first:

- I would like to see the authors specifically address the issue of under-reporting or cryptic transmission. The analysis appears to be quite dependent on the assumption that all transmission events are seen. That may be a reasonable assumption for this particular outbreak, but is likely not the case for larger outbreaks or non-Ebola outbreaks.

Response: *We agree with the reviewer that an assumption that all cases are detected is likely to be reasonable for this particular outbreak, during which intensive contact tracing and community testing took place. However, we also agree that for larger outbreaks (or non-Ebola outbreaks) underreporting can be a substantial challenge. The reviewer's comment has therefore prompted us to consider how our modelling framework can be extended to account for underreporting.*

Specifically, we have conducted an entirely new analysis in which we analyse data from a larger Ebola outbreak (please see below). For that outbreak, since underreporting is more likely to have occurred than in the 2017 Ebola outbreak in Likati Health Zone, we have reproduced our results but instead accounted for the possibility that cases were missed (see Supplementary Analysis 3 and Fig S4). To do this, we first calculated the relative probability of a missed case occurring on each day of the outbreak based on the cases that were observed. We then sampled unreported cases from that probability distribution, and estimated the value of R (in the absence of the ERT) and the risk of withdrawing the ERT using the new "dataset" in which the unreported cases are included. We then repeated this procedure a large number of times (10,000 times), enabling us to calculate the risk of withdrawing the ERT across a wide range of possible dates on which cases occurred but were not reported. As expected, when underreporting is accounted for, the risk of withdrawing the ERT is higher (Fig S4B). This analysis demonstrates how underreporting can be accounted for in quantitative estimates of the ERT withdrawal risk.

We note that, when the risk of withdrawing the ERT falls to very low levels (so that policy advisors might consider withdrawing the ERT), the precise number of missed cases may not affect the practical conclusions drawn from the analysis as much as might be expected. For example, in Fig S4, if the ERT is withdrawn as soon as the risk of withdrawing the ERT falls below 0.01, then if two cases are missed the removal date would be 24th July 2018 whereas if five cases are missed then the equivalent removal date would only be five days later (29th July 2018). We note this alongside this additional analysis in Supplementary Analysis 3. We also highlight this new analysis in the main text of the revised manuscript (lines 743-752 and 854-861).

- the authors discuss the assumption that the number of cases per day is Poisson distributed as a limitation. This seems a strong assumption that should be addressed here. It would help to see Figures 2 and 3 with alternative assumptions (I presume that alternative assumptions will impact both the estimate of R (Figure 2a) and the probability of more cases upon removal).

Response: *We agree with the reviewer that the assumption that the number of cases per day is drawn from a Poisson distribution is a restrictive one (albeit one that is common when renewal equation models are employed; see e.g. the renewal equation underlying the R software package EpiEstim – developed by Cori et al. and used worldwide to estimate the time-varying reproduction number during the COVID-19 pandemic).*

In our revised manuscript, we have therefore conducted a new analysis (Supplementary Analysis 1) in which we relax this assumption. Specifically, we have considered scenarios in which the number of cases per day is instead drawn from a negative binomial distribution, with values of the dispersion parameter, k , that are either less than one (Fig S2 – blue lines) or greater than one (Fig S2 – red lines). A lower value of k corresponds to a higher chance of superspreading events occurring. As the reviewer suggests, this leads to different estimates of R and the risk of removing the ERT. However, in both of these analyses, our main conclusions remained unchanged: in the absence of the ERT, more cases would have been likely, and the ERT was removed only when it was safe to do so (in Fig S2B, the risk of withdrawing the ERT was estimated to be 0.017 at the actual time that the ERT was removed when a value of $k=0.2$ was used, and 0.010 when a value of $k=10$ was used). As noted in our response to Reviewer 2, in terms of practical decision making, the risk of removing the ERT was similar for different values of k when the risk of removing the ERT reached low levels (since these are the levels at which the ERT is likely to be removed, this demonstrates the robustness of our practically relevant results to the assumed value of k). We have highlighted this new analysis in the main text (lines 720-733 and 840-853).

- I would like to see an analysis of the sensitivity of the results to e.g. a missed case at the time of the decision to remove the intervention. Especially given the small size of this outbreak, I would think that missing a single case might be impactful (but would be happy to be proven wrong). This gets at the interaction between this particular decision framework and the strength of surveillance. For example, I could imagine a result where

this is a very robust decision framework IF one is very confident in the surveillance (numbers of cases and timing of cases), but one might want to move to a more conservative risk threshold if you're less confident of the surveillance.

Response: *Thank you for encouraging us to consider the impact of missed cases on estimated risks of withdrawing the ERT. As noted above, we have now conducted a further analysis to demonstrate how missed cases can be accounted for in our modelling framework (Supplementary Analysis 3 and Fig S4). By assuming that more cases are missed, a policy maker is able to make more conservative decisions about when to remove interventions such as the ERT.*

*In terms of the effect of a missed case at the time that the intervention is due to be removed, we agree that this would be impactful (it would lead to a higher risk of withdrawing the ERT than estimated under an assumption that no cases were missed). However, the results from *any* decision-making framework for deciding when to remove the ERT would be drastically changed by the addition of a case at the time of the decision to remove the intervention. For example, under the WHO's current decision-making guideline, which involves withdrawal of the ERT following a period of 42 days without cases (following the death or safe burial of the previous case), then an additional case at the time of ERT withdrawal would delay withdrawal by at least 42 days.*

Furthermore, in Fig S4A, it can be seen that if a missed case was infected by one of the reported cases in the dataset shown in Fig S3A, then it is highly unlikely that the missed case occurred around the time that the ERT was removed. Consequently, after substantial consideration, we think that the results shown in Fig S4B are a more appropriate representation of the implications of under-reporting than results generated by artificially inserting a missed case at the time of the decision to remove the ERT.

We have highlighted our new analysis exploring the effects of under-reporting in the Results (lines 743-752) and Discussion (lines 854-861) of the revised manuscript.

With respect to the second:

- I am concerned that such a small outbreak may be idiosyncratic and misleading. I'd recommend that the authors consider either a simulation study based on rates estimated here (perhaps with alternative assumptions about the offspring distribution for the branching process), or an analysis of a larger outbreak where the impact of the intervention is less clear (e.g. cases continue after the intervention). I would prefer the latter as it would help to argue for the generality of this approach.

Response: *Thank you for encouraging us to consider applying our modelling approach to data from a larger outbreak. We have now done this, considering an Ebola outbreak from Équateur Province in the Democratic Republic of the Congo in which 54 cases arose (Supplementary Text 2 and Fig S3). In that outbreak, some cases occurred after the arrival of the ERT. In addition, the estimated value of R in the absence of the ERT was higher for that outbreak than in the outbreak underlying our main analysis. We*

hope that this is sufficient to demonstrate the general nature of our approach – and that our modelling framework is useful to inform the relaxation of interventions in a wide range of future outbreaks.

- a challenge of the simulation approach is that the lack of onward cases after the introduction of the ERT implies that there is large uncertainty in the effectiveness of the ERT; e.g. did it reduce R to 0, less than 1, or greater than 1 and fadeout was a random occurrence? The statement on L147-149 implies that the authors assume that no new case arise after the ERT, again this seems a strong and limiting assumption. The serial interval distribution that the authors assume implies that there is at least a non-trivial probability that transmissions events at the approximate time of the ERT might not even be observed for several weeks. It is interesting to note that the longer the ERT, the more certainty one has in the absence of new transmission; but again, this interacts with the assumption about the effectiveness of the ERT. One might not feel confident in the assumption of effectiveness 1 day after the ERT, but might feel very confident by the 5th of April.

Response: *Thank you very much for the opportunity to clarify these points. We apologise that the statement on lines 147-149 of our original submission was unclear. The key point here is that, to calculate the probability that no cases occur from day t onwards (i.e. no cases on day t , $t+1$, $t+2$ etc), we first calculate the probability that no cases arise on day t . Then we calculate the probability that no cases occur on day $t+1$, given that no new cases arose on day t (i.e. setting $I_t = 0$). Then we calculate the probability that no cases occur on day $t+2$, given that no new cases arose on days t or $t+1$ (i.e. setting $I_t = 0$ and $I_{t+1} = 0$). And so on. This allows us to calculate the overall probability that no future cases will occur as seen on day t . Crucially, we do not assume that no cases can occur after day t , but instead calculate the probability that no cases occur from day t onwards. We have revised the wording in the text accordingly (lines 931-933).*

The risk of withdrawing the ERT as estimated each day accounts for any cases observed up to (and including) that day. While no new cases arose after the ERT arrived in the dataset from the 2017 EVD outbreak in Likati Health Zone, this was not true in the analysis of the larger outbreak considered in Supplementary Analyses 2 and 3. As such, it is crucial that our method can handle the possibility of cases occurring between the arrival of the ERT and day t (as indeed it can; we do not assume that no cases occur after the ERT is deployed).

We agree with the reviewer that there will always be substantial uncertainty in the effectiveness of the ERT soon after it is deployed. Our conclusion that our results suggest that the ERT was effective in the 2017 outbreak is based on the fact that no further cases went on to occur (without the ERT, we estimated that the risk of additional cases following 15th May 2017 would have been 97% - corresponding to only a 3% chance of no further cases occurring). In other words, this conclusion was obtained only once it was known that no further cases occurred. In contrast, our real-time estimates of

the risk of withdrawing the ERT on day t are based only on cases observed up to (and including) day t ; those estimates can therefore be obtained in real-time.

The reviewer is correct that it is possible that some infections may have occurred, but that the infected individuals may not yet have developed symptoms, on the date of ERT withdrawal. However, by using observations of symptomatic cases and the serial interval distribution as inputs to our modelling framework, this possibility is accounted for.

In the revised manuscript, in addition to clarifying the statement originally on lines 147-149 (lines 931-933 of the revised manuscript), we have explained in the Discussion that the risk of withdrawing the ERT can be inferred in real-time, but that the effectiveness of the ERT could only be assessed afterwards (when no further cases had occurred); please see lines 774-778 of the revised manuscript.

Finally, it strikes me that the assumption of an explicit distribution of serial interval is a very strong assumption that limits this analysis to other Ebola outbreaks, or outbreaks with a well defined serial interval. If the serial interval is not well known, it is not clear that it could be well estimated from such a small outbreak. However, for a larger outbreak, this could also be estimated and the entire analysis self contained to the specific outbreak at hand. I realize that practical decision-making often requires making such assumptions, but the dependence of these results on this assumption and the limitation of the direct application of this framework to settings with such a well characterized distribution should be addressed here.

Response: *Thank you for pointing this out. We agree, and have now considered a scenario in the revised manuscript in which, for the larger Ebola outbreak, we estimate the serial interval distribution using data from that outbreak. We also compare the resulting estimates of both R and the risk of withdrawing the ERT using either the outbreak-specific serial interval or the original general Ebola serial interval parameterisation (Figs S3D,E).*

As noted in our response to Reviewer 2, we found that differences between the two serial interval distributions affected the precise estimated values of both R and the risk of withdrawing the ERT (Fig S3D,E), although the general pattern remained unchanged (the longer the period elapsed following the end of the outbreak, the lower the risk of withdrawing the ERT). Differences in the exact quantitative estimates highlight the importance of using outbreak-specific serial interval estimates when they are available, as we have now highlighted at the end of Supplementary Analysis 2.

Ultimately, I think the authors have presented a very interesting and clean framework for addressing the problem of when to declare the end of NPIs for a small Ebola outbreak where NPIs have been highly effective. That is an achievement by itself, but not one that I think rises to the level of generality that I would expect of Nature Communications. I think, with substantial further analysis, this work could be broadened to illustrate the extent of its generalizability.

Response: *Thank you very much for your positive comments. We hope that the new analyses described above, including: i) considering the effect of superspreading events; ii) analysing data from a larger EVD outbreak (including inferring the serial interval using data from that outbreak) and iii) demonstrating how underreporting can be accounted for in our modelling framework; have addressed your concerns. We hope that these further analyses are sufficient to illustrate the generalisable nature of our approach.*

Reviewer 4

This is a strong manuscript and novel use of modeling to show a positive public health impact. The writing was clear and i don't see need for edits.

Response: *Thank you for these positive comments. We are glad that you enjoyed reading our manuscript.*

REVIEWERS' COMMENTS

Reviewer #1 (Remarks to the Author):

I have no further comments or questions. I think the authors did a great job of responding to the reviewers. In all cases where I felt I needed more information, that information was provided. I appreciate the "conversation" with the authors regarding this manuscript, particularly as it relates to estimating R.

Reviewer #2 (Remarks to the Author):

The authors have addressed my comments, incorporated the suggestions put forth, and revised the manuscript.

Reviewer #2 (Remarks on code availability):

The code is reproducible.

Reviewer #3 (Remarks to the Author):

In general I am happy with the changes made by the authors and am happy to recommend the paper for publication. One point of clarification should be made to the analysis in Supplementary Analysis 2. Here the authors estimated the serial interval from observed contact tracing data. They indicate that this included 41 realized serial intervals. But by the time of their first "real time" estimate of the risk of removing the ERT this number was smaller (if I am reading correctly). This implies that the authors uses all available data to estimate the serial intervals, including serial intervals that had not yet been observed as of 6 August. If that is the case, then the estimate overstates the sample size and confidence in the estimate. If it is not the case, and the authors used, on each date, only those serial interval data that would have been in-hand for a real-time estimate, then they should make that statement clearer in the supplement.

Reviewer 1

I have no further comments or questions. I think the authors did a great job of responding to the reviewers. In all cases where I felt I needed more information, that information was provided. I appreciate the "conversation" with the authors regarding this manuscript, particularly as it relates to estimating R.

Reviewer 2

The authors have addressed my comments, incorporated the suggestions put forth, and revised the manuscript.

The code is reproducible.

Response: *Thanks again to Reviewers 1 and 2 for their comments and suggestions that helped us to improve our manuscript.*

Reviewer 3

In general I am happy with the changes made by the authors and am happy to recommend the paper for publication. One point of clarification should be made to the analysis in Supplementary Analysis 2. Here the authors estimated the serial interval from observed contact tracing data. They indicate that this included 41 realized serial intervals. But by the time of their first "real time" estimate of the risk of removing the ERT this number was smaller (if I am reading correctly). This implies that the authors use all available data to estimate the serial intervals, including serial intervals that had not yet been observed as of 6 August. If that is the case, then the estimate overstates the sample size and confidence in the estimate. If it is not the case, and the authors used, on each date, only those serial interval data that would have been in-hand for a real-time estimate, then they should make that statement clearer in the supplement.

Response: *The reviewer is correct in their interpretation of our use of serial interval data in Supplementary Analysis 2. The goal of using a different serial interval distribution in this analysis (as opposed to the commonly used EVD serial interval estimate in the main text) is to demonstrate that the assumed serial interval distribution affects estimates of the risk of withdrawing the ERT. We have added a statement to the Supplementary Information noting that this analysis uses all observed serial intervals from the outbreak in question when estimating the serial interval distribution: "In particular, following contact tracing, 41 realised serial intervals were recorded (these data represent all available observations of the serial interval recorded throughout the 2018 Équateur Province outbreak)."*

Thanks again to the reviewer for their help improving our manuscript.